# Mini-Review on the Possible Interconnections between the Gut-Brain Axis and the Infertility-Related Neuropsychiatric Comorbidities

**DOI:** 10.3390/brainsci10060384

**Published:** 2020-06-17

**Authors:** Gabriela Simionescu, Ovidiu-Dumitru Ilie, Alin Ciobica, Bogdan Doroftei, Radu Maftei, Delia Grab, Jack McKenna, Nitasha Dhunna, Ioannis Mavroudis, Emil Anton

**Affiliations:** 1Department of Mother and Child Medicine, Faculty of Medicine, University of Medicine and Pharmacy “Grigore T. Popa”, University Street, No 16, 700115 Iasi, Romania; gabi.ginecologie@gmail.com (G.S.); delianicolaiciuc@yahoo.com (D.G.); emil.anton@yahoo.com (E.A.); 2Clinical Hospital of Obstetrics and Gynecology “Cuza Voda”, Cuza Voda Street, No 34, 700038 Iasi, Romania; dr.radu.maftei@gmail.com; 3Origyn Fertility Center, Palace Street, No 3C, 700032 Iasi, Romania; 4Department of Research, Faculty of Biology, “Alexandru Ioan Cuza” University, Carol I Avenue, No 11, 700505 Iasi, Romania; ovidiuilie90@yahoo.com (O.-D.I.); alin.ciobica@uaic.ro (A.C.); 5Department of Morphostructural Sciences, Faculty of Medicine, University of Medicine and Pharmacy “Grigore. T. Popa” Iasi, University Street, No 16, 700115 Iasi, Romania; 6York Hospital, Wigginton Road Clifton, York YO31 8HE, UK; jackscloudinthesky@me.com; 7Mid Yorkshrie Hospitals NHS Trust, Pinderfields Hospital, Wakefield WF1 4DG, UK; n.dhunna@hotmail.co.uk; 8Leeds Teaching Hospitals NHS Trust, Great George St, Leeds LS1 3EX, UK; ioannis.mavroudis@nhs.net; 9Laboratory of Neuropathology and Electron Microscopy, School of Medicine, Aristotle University of Thessaloniki, 541 24 Thessaloniki, Greece

**Keywords:** microbiome, gastrointestinal, neuropsychiatry, infertility

## Abstract

Both the gut-brain axis (GBA) and the hypothalamic–pituitary–adrenal (HPA) axis remain an intriguing yet obscure network with a strong influence over other systems of organs. Recent reports have sought to describe the multitude of harmful stressors that may impact the HPA axis along with the interconnections between these. This has improved our knowledge of how the underlying mechanisms working to establish homeostasis are affected. A disruption to the HPA axis can amplify the chances of gastrointestinal deficiencies, whilst also increasing the risk of a wide spectrum of neuropsychiatric disorders. Thus, the influence of microorganisms found throughout the digestive tract possess the ability to affect both physiology and behaviour by triggering responses, which may be unfavourable. This is sometimes the case in of infertility. Numerous supplements have been formulated with the intention of rebalancing the gut microflora. Accordingly, the gut flora may alter the pharmacokinetics of drugs used as part of fertility treatments, potentially exacerbating the predisposition for various neurological disorders, regardless of the age and gender.

## 1. The Current Understanding of the “Second Brain”

Following early studies into the relationship between humans and bacteria, attained through a series of microscopic observations [1], came the largest research project dedicated to all-commensal, symbiotic and pathogenic entities three and a half centuries later [2]. In 2008, this Human Microbiome Project (HMP) was launched. The main objective(s) of the initial, or “Jumpstart” phase, were to develop new algorithms (libraries with reference sequences), technologies and tools dedicated to the assessment of the intestinal microflora [3].

It has been concluded that all the microorganisms which colonise our body are grouped into four major microbial categories. The gastrointestinal microbiota (GM) exceeds the oral, urogenital and skin microbiota, and even outnumber the total number of the human somatic and germ cells by a factor of 10 [4]. The microbes are spread throughout the entire gastrointestinal tract [5,6], with anaerobic microorganisms sub-divided into three enterotypes: *Ruminococcus*, *Prevotella* and *Bacteroides* [7]. The most abundant bacterial collections are the *Firmicutes*, *Bacteroidetes*, *Actinobacteria* and *Verrucomicrobia* phyla [8], collectively unifying over 1000 species that have been cultured and analysed phylogenetically [9].

The Integrative Human Microbiome Project (iHMP) then centred its focus on the influence of the gut’s flora on transient episodes—more precisely, between eubiosis and dysbiosis. There are three aspects to the second research phase; each aspect is considered in the context of the commensal, enteric bacteria. These phases were (I) pregnancy, delivery method and premature births; (II) irritable bowel syndrome (IBS) and its potential triggers; and finally, (III) the influence of stressors in the pathophysiology of prediabetes [10].

The gastrointestinal microbiota is composed of distinct cell types [11,12,13,14] which fulfil key roles [15] in order to prevent a dysbacteriosis. These different cell types initiate specific responses, defining its crucial role in maintaining the integrity of the neurohormonal axes [16].

Even if the concept regarding the two-way path between the intestinal flora and the brain is generally accepted, these relations are still insufficiently understood. The gut–brain axis (GBA) is a dense network which unites a number of fundamental physiological pathways, such as the central nervous system (CNS), the neuro-endocrine and -immune systems as well as the sympathetic and parasympathetic components of the autonomic nervous system (ANS) and the enteric nervous system (ENS). The hypothalamic–pituitary–adrenal (HPA) axis also plays a pivotal role alongside the GBA in many stress-related disorders [17].

Recent findings support the notion of a “personalised microbiome”, with an inter-individual variation through a series of endo- and exogenous factors. The commensal microbiota are influenced by interactions with pathogens [18] along with dietary differences [19], and prolonged exposure to medications [20]. The myriad of influencing factors contribute to the formation of the individual human virome [21], as well as the ‘gut resistome’ or antibiotic resistance [22]. Other factors that can affect the micro-environment include a lack of physical exercise [23], as well as the influence of heritable components [24], and the culmination of these factors promotes transitions of bacteria in different sites along the gut [25].

Apart from the metabolism of the gut microbiota, characterised by a wide variety of metabolites involved in a host’s eubiosis [26,27] based on the exogenous supply, these microbial associations as well participate in shaping a newborn’s microbiota [28].

## 2. Disruptive Factors in Enteric Eubiosis and the Influence on Colonisation in Neonates

One of the earliest interactions of the foetus with the maternal urogenital microbiota take place once the foeus passes into the birth channel [28].

The colonisation process could actually be initiated in utero; Collado et al. [29] identified that *Proteobacteria* is the most prevalent phylum in both the placenta as well as the amniotic fluid. Data obtained following the analysis of the meconium suggest a mother–foetal transfer, with infants’ microbiota being similar to that found in the colostrum after almost one week. The neonatal microbial communities are influenced by several processes such as preterm deliveries along with the method of delivery.

Aagaard et al. [30] have highlighted the existence of a temporary niche formed during pregnancy, which unites four phyla: *Firmicutes*, *Tenericutes*, *Proteobacteria*, *Bacteroidetes* and the *Fusobacteria* genus. While *Bifidobacterium*, *Lactobacillus*, *Bacteroides* and *Clostridium* are passed via the placenta [31,32], Lauder et al. [33] concluded that there are no significant differences between the number of copies following a q-PCR analysis between the placental strains and the negative controls.

However, whether or not the placenta possesses beneficial microorganisms is still under question, mainly because some recent evidences supports the notion of favourable conditions for certain pathogen proliferation—in particular, *Group B streptococcus* [34].

Stout et al. [35] established that 27% of the basal plates of placentas possess intracellular bacteria which is, therefore, a possible route for intra-uterine colonisation. The finding of placental intracellular bacteria was found in 54% of the studied cohort who had a spontaneous preterm delivery, and in only 26% of term-spontaneous deliveries. There were no major differences in the predisposition for intra-amniotic infections or *Group B Streptococcus* in preterm births.

Intrauterine infections are known to be a cause of both spontaneous preterm labour (PTL) and/or preterm prolonged rupture of membranes (PPROM) [36,37]. As the bacterial DNA has been detected in 70% from all the placental tissues, the authors concluded that the placental membranes possess bacteria, but it is not a cause of preterm labour or PPROM [38] following a caesarean section (C-section). On the other hand, no signatures of bacterial DNA have been detected compared with term vaginal deliveries, the positivity being around 50% [39].

The gestational age of an infant can correlate to the diversity seen in the commensal bacteria that are acquired by the infant, which is suggestive of prenatal influences [40,41]. Interestingly, Hu et al. [42] have concluded that the meconium unites microbial strains, arguing that the mode of birth does not influence the microbial diversity. However, the microbial composition of the meconium was significantly influenced by the maternal diabetes status. 

It is intriguing that one of the microbes involved in the metabolism of levodopa in patients with Parkinson’s disease has been identified in meconium samples. With a rate of 1 to 5 samples, *Enterococcus faecalis* has been identified in almost 80% of all the samples after the meconium has passed within the first two hours [43].

Hansen et al. [44] evaluated the microbiota contained within the meconium, and they found that bacteria was detectable in two-thirds of the meconium samples through the use of fluorescence in situ hybridization (FISH) and 7% by standard polymerase chain reaction (PCR), while a significant percentage of sterile samples have been defined by a minimum inhibitory concentration (MIC).

*Enterococcus*, *Streptococcus*, *Staphylococcus*, or *Propionibacterium* were the predominant strains in the umbilical blood cord [43], while in the amniotic fluid the bacterial composition was dominated by species such as *Sneathia sanguinegens*, *Leptotrichia amnionii* and an uncharacterised bacteria [45]. Shao et al. [46] have identified that the bacterial composition can be influenced by delivery method, as they demonstrated a disruption to the transmission of the maternal *Bacteroides* in caesarean sections, with C-sections proving to detriment the *Enterococcus*, *Enterobacter* and *Klebsiella* species.

In addition, preterm infants often receive treatment with antibiotics in order to prevent possible infections, but a recent research article has demonstrated the subsequent existence of resistome as a result of prolonged exposure to various drugs [47], suggesting that an infant’s commensal microbiome will be impacted by these antibiotics.

## 3. Enteric Microbial Variations in Childhood: The Heritable and Social Components

Turnbaugh et al. [48] revealed that even monozygotic (MZ) pairs have distinct signatures of commensal microbiota. The stool samples collected were compared to 1095 bacterial communities that are commonly found in the gut and other body habitats, from related and unrelated individuals. In over one million bacterial reads, the α-diversity indicated approximately 800 following the analysis of the hypervariable V2 region.

Goodrich et al. [49] have reproduced an association between the lactic bacteria belonging to the *Bifidobacterium*, which is usually heritable between the UK twins, and the LCT gene locus, being responsible for the hydrolysation of lactose in the upper GI tract. 

The faecal samples collected from mono (MZ)- and dizygotic (DZ) twin individuals from the United States and South Korea have revealed the existence of a unique microbiome. Based on the sequences obtained following the analysis of the bacterial V2 region, Lee et al. [50] have concluded that this variation seems to be the result of a combination between some temporal and spatial variables.

This hypothesis also applies to a much smaller degree in brothers. In a study conducted by Schloss et al. [51], a metagenomic shotgun analysis of 16S rRNA’s V3-V5 region has been conducted with the aim of distinguishing the microbial communities of each family member having as reference individuals which live in the same geographic area. *Bifidobacterium* and *Escherichia* have been the most dominant strains encountered in all siblings, with the mention that the microbiota of the two-year-old was more similar to her weaned siblings. Twelve operational taxonomic units (OTUs) have been identified within the family, from which four were location specific, belonging to the genus *Bacteroides* and *Subdoligranulum* and family *Lachnospiraceae*.

Recently, Kato et al. [52] revealed the presence of the CC genotype in 1068 Japense adults at rs4988235 and the GG at rs182549, in addition to those previously reported (rs145946881, rs41380347, rs41525747 and rs869051967). They found that there was positive correlation between the CC genotype and a low abundance of *Bifidobacterium* [53,54]. C/T(-13910) has been mainly reported as the predominant lactase locus in Europeans, while G/A(-22018) in Japanese–Brazilian and Chinese populations [55]. In addition, bathtub water has proved to be a potential vehicle for the bacterial transfer and it is not strictly a mother-to-infant axis.

Odamaki et al. [56] enrolled 21 Japanese individuals from five families and, after the isolation of the faecal and bathtub samples, *Bifidobacterium longum* was shown to be the most abundant microorganism exchanged between the members, compared with those which do not adopt this tradition. A comparative study conducted by the same author demonstrated that *Bifidobacterium longum* subsp. *longum* is present throughout the entire life, regardless of age. Their results suggest that some bacteria are distributed across family members [57].

Laursen et al. [58] have evaluated how early infections, having older brothers or pets could disrupt the normal colonisation of the gut. David Strachan’s hygiene hypothesis has been certified in the present study, with the presence of older siblings being positively correlated with the bacterial diversity and richness of *Firmicutes* and *Bacteroidetes* or with *Faecalibacterium prausnitzii* [59]. On the other hand, pets or early infections had less contribution towards the gut flora, without any significant data correlation.

Dill-McFarland et al. [60] have emphasised in his study an attribute acquired as social human beings. The analysis of the faecal samples collected from 177 individuals, from which 94 were spouses and 83 were siblings, revealed that their taxa is more similar and diverse when compared with those of related or unrelated individuals, with the cause–effect relating to dietary habits.

The faecal samples collected over an interval of two years has showed that are no major fluctuations within these communities, being stable throughout the entire study. The only minor difference was in the case of one person after an intervention that required medication, and no foreign or major change regarding species density has been reported [61].

## 4. How Is the HPA Axis Influenced?

A reduction in the host’s innate eubiosis triggers a pro-inflammatory cascade [62]. If this state is prolonged it may lead to gastrointestinal disorders [63,64], as well as neurodegenerative [65] or neuropsychiatric disorders [66].

In such cases, the HPA axis exerts an antagonistic effect upon the organism. It has been shown that the patients with a major depressive disorder (MDD) have high serum levels of cortisol [67] and reduced levels of oxytocin [68]. The results obtained in another study, conducted with a similar design to the previous one, provides additional evidence and further consolidates this strong correlation between the brain and the digestive tract [69].

There is a lot of controversy regarding the interconnections between the neurological and gastrointestinal disorders [63,66] and even if these disturbances of the central nervous system (CNS) are irremediable, at least the symptoms can be reduced by enhancing the GM.

A number of conventional alternatives have been developed in recent years [70,71], presently being considered the most powerful vehicles for the acquisition of the beneficial microorganisms intended for the reconstruction of the GM (Table 1).

It has been suggested that intestinal microflora may reduce or even inhibit the treatment for infertility [82] and, in parallel, gradually promote neuropsychiatric disorders (Table 2).

It must be taken into consideration that in Table 2, we have focused only on those drugs usually administered for infertility with a known potential for a sside effect that include the promotion of a psychiatric disorder.

Unfortunately, very little is known regarding the side effect profile of infertility medication.

Thus, it can be concluded on the basis of the studies summarised in Table 2 that infertility drugs indeed exert antagonistic effects upon the neurohormonal axis by disrupting its normal functionality.

It is difficult for patients and clinicians to figure out which responses are psychological and which are caused by medication, but it is vital to identify the causes in order to determine the future measures.

## 5. How Infertility and Associated Drugs Disrupt the HPA Axis?

Infertility can have profound consequences on a person’s psychology, often through the perception of losing control on one’s life [94]. The issue of infertility can become centric to a relationship with anger and confusion replacing reason [95]. This is because an adults’ progress and identity often resides with the desire to conceive [96]. Cousineau [97] extensively reviewed all the aspects surrounding the issues in relation to the cultural and social effects of infertility, along with the influence on marital status and decision making and the relevant psychological support. Infertility treatment puts a great deal of stress on a couple and as this can culminate in an attitude of resignation, and the aspiration to have a child is replaced by adoption or being child free [98].

Many couples find it difficult to adapt to this new trajectory, and often find it hard to acquire a new vision beyond this temporary crisis [99]. They must often make radical lifestyle adjustments, such as re-evaluating decisions surrounding career options, with other important aspirations often postponed [100]. Aside from the individual lifestyle upheaval, one must adapt to a rigorous medication program [101].

Infertility should not be viewed as a major impediment, but rather as an unplanned event. The literature highlights a broad array of causes for infertility. The ones that have been most emphasised lately are polycystic ovarian syndrome (PCOS) [102,103], hypothalamic dysfunctions [104], premature ovarian failure (POF) [105] and endometriosis [106]. However, rather than focussing on the organic aetiology behind infertility, we have decided to detail the associated neuropsychiatric comorbidities, given the fact that women are more prone to mixed anxiety–depressive disorders (MAAD) than men. This is why, in Table 3, we have summarised all the studies conducted between 2010–2020, focussing on large cohorts (≥1000 patients per sample).

## 6. Conclusions

The evidence presented in this manuscript suggests that the gut microflora has a profound and complex influence on the psychological profile of each individual. Moreover, changes in this integrative system may serve as a bridge to upcoming CNS disorders, whilst also having the potential to provoke gastrointestinal deficiencies in an early stage. Regarding the infertility medication and the overall “disease”, this topic remains debatable, mainly because the number of studies is limited, but it is clear that it may disrupt the integrity of the GBA–HPA axes. However, an occurring dysbacteriosis not only gradually alters homeostasis, but also amplifies the chances to block entirely the effect of any infertility drug. On the other hand, the techniques dedicated to the restoration of the microbial communities have undergone a fulminant ascension lately but, like any therapy, some disorders have been omitted for unknown reasons, which is why additional studies will further aid our understanding.

## Figures and Tables

**Table 1 brainsci-10-00384-t001:** The modulatory effects following the administration of probiotics in the regulation of the GBA–HPA axes.

Number of Patients	Main Observations	Reference
33 autistic children	After the administration of a probiotic for 21 days which contained three species of *Lactobacillus* (*acidophilus, casei* and *delbruecki*), two of *Bifidobacterium* (*longum*, *bifidum*), 88% of individuals reported a significant reduction in ATEC, 52% for constipation and 48% for diarrhoea	[72]
10 autistic children and controls and 9 siblings	After the administration of a probiotic three times a day for four months which comprised of three strains, including: *Lactobacillus*, *Bifidobacterium* and *Streptococcus*, increased levels of *Bacteroidetes* and *Firmicutes* were reported, concomitant with the normalisation of *Lactobacillus* and *Bifidobacterium* spp. ratio	[73]
75 infants	After the administration in the first months of life of a probiotic which contained one species of *Lactobacillus* (*rhamnosus*) for half a year, at the age of 13, approximately 17.1% of the children from the placebo group have been diagnosed with ASD or ADHD and none from the probiotic group	[74]
11 autistic children	Compared the administration of a probiotic, containing three species of *Lactobacillus* (*acidophilus*, *bulgaricus* and *bifidum*), to a cohort receiving 500 mg of Vancomycin four times per day for two months. There was a significant improvement in the general health as assessed by a clinical psychologist	[75]
30 autistic children ranging from 5 to 9 years	Following the administration of a probiotic containing two species of *Lactobacillus* (*acidophilus* and *rhamnosus*) and one of *Bifidobacterium* (*longum*) for three months, PCR-based methods revealed increased levels of *Bifidobacterium* and *Lactobacillus*. This was subsequently associated with a reduction in the body weight, autistic symptomatology and of gastrointestinal symptoms based on the ATEC and 6-GSI questionnaires, respectively	[76]
62 autistic children	Following the administration of a probiotic which contained one species of *Lactobacillus* (*plantarum* WCSF1) for three months, the placebo group was found to have a higher prevalence for antisocial behaviours, anxiety and communication deficits than the probiotic group	[77]
12-year-old autistic boy with severe cognitive disability	Following the administration of a probiotic (VSL#3) for one month—containing three species of *Bifidobacterium* (*breve*, *longum* and *infantis*), five of *Lactobacillus* (*acidophilus*, *plantarum*, *casei*, *bulgaricus*, *delbrueckii* subsp) and two of *Streptococcus* (*thermophilus*, *salivarius* subsp.), with an additional month of follow-up—the severity of the abdominal symptoms had significantly reduced. This was followed by an overall improvement of the core symptomatic panel	[78]
65 schizophrenic patients	After the administration of one species of *Lactobacillus* (*rhamnosus* strain GG) and one of *Bifidobacterium* (*animalis* subsp. lactis strain Bb12) for three and a half months, the patients no longer manifested any specific symptom	[79]
31 chronic schizophrenic patients and 27 placebo	After the administration for three and a half months of one species of *Lactobacillus* (*rhamnosus* strain GG) and one of *Bifidobacterium* (*animalis* subsp. lactis strain Bb12), a bowel movement improvement was observed, which has been positively correlated with the reduction of a series of specific biomarkers	[80]
56 schizophrenic patients	After the administration of an adjuvant probiotic which contained one species of *Candida* (*albicans*) and one of *Saccharomyces* (*cerevisiae*) over a four-month period, it was found that there was a reduction in *Candida* IgG only in men, associated with a better functioning of the GM, and with no significant differences for *Saccharomyces* in both groups	[81]

ATEC—Autism Treatment Evaluation Checklist; ASD—Autism Spectrum Disorder; ADHD—Attention Deficit Hyperactivity Disorder; 6-GSI—six-item Gastrointestinal Severity Index.

**Table 2 brainsci-10-00384-t002:** Infertility drugs with a known effect to induce pronounced mood fluctuations as a side effect.

Type of Drug	Number of Patients	Main Observations	Reference
CC	50 patients (25 couples)	CC have exacerbated symptoms of PMS in 9 out of 14 women only (e.g., irritability and mood swings)	[83]
CC	1 male patient	The patient in this study was diagnosed with oligoteratospermia and had received CC. The treatment culminated in depressive symptoms for five consecutive days. After the treatment was discontinued, it took an additional seven days until he made a full recovery	[84]
CC and hMG	454 (139 women who had not previously taken any drug and 315 who had previously received medication)	This cross-sectional, self-reported study concluded that both CC (*n* = 162) and hMG (*n* = 153) act as agonists and could trigger disorders such as depression according to the STAI through a disturbance of the HPA axis. Significant differences were noted between the groups, with those women taking either CC or hMG reporting a higher incidence of psychological effects	[85]
OC	34 women (17 COC and 17 placebo)	During the seven-day study period, the COC users displayed more depressive symptoms when compared to the placebo cohort according to the CD scale. This was highlighted by a specific reactivity at the level of the insular cortex, respectively, the first one-third and the lowest portion of the frontal lobe through fMRI both before as well as during the treatment	[86]
OC	76 women (38 OC and 38 placebo)	A significant percentage (77%) of the total adolescent cohort had one side effect manifested. Interestingly, the number and type of side effects were identical in both the OC and placebo groups after the completion of CES-D	[87]
HC	1,061,997 Danish women	Compared with the relative low risk once with aging, adolescents have been more predisposed to the subsequent usage of antidepressants following the administration of HC	[88]
HC	2532 women (232 oestrogen–progestin, 58 progestin only and 948 with no treatment)	The use of combined hormone contraception has been higher in Caucasians with MDD, while those on progesterone monotherapy displayed more hypersomnia, weight gain and a relatively worse physical functioning. Those with the COC were singnificantly less depressed than those in the other two groups according to the QIDS score	[89]
HC	75,528 postpartum women	From the total, 7.8% were prescribed antidepressants, while 5% have been diagnosed with depression, percentages which differ depending on the type of the hormonal contraception. It should be noted that the women had previously served in the US army	[90]
HC	815,662 Swedish women	The high CI (95%) OR indicate a strong correlation between psychotropic drug usage among adolescents compared with the older women	[91]
CC versus AIs for PCOS as well as gonadotropins versus CC versus AIs in patients with an unknown cause of infertility	3258 patients (1650 women and 1608 men)	In women who were not previously taking any antidepressants, MD did not negatively influence the fertility chances, but instead slightly increased the likelihood of pregnancy. However, in 90 of the women who had taken antidepressants previously, there was an increased risk of miscarriage, while in men, active MD reduced the likelihood for conception	[92]
GnRHa	29 women (agonist)	GnRHa has been positively correlated with a pronounced depression-like symptomatology, analogue with anxiety, but this have been considered an overlap of the pre-existing condition in euthymic participants according to the HAM-A, HAM-D and VAS-A and VAS-D	[93]

CC—clomiphene citrate; PMS—premenstrual syndrome; hMG—human menopausal gonadotropin; HPG—hypothalamic–pituitary–gonadal; STAI—State-Trait Anxiety Inventory; OC—oral contraception; CD—Cyclicity Diagnoser; fMRI—Functional Magnetic Resonance Imaging; COC—combined oral contraception; CES-D—Center for Epidemiologic Studies Depression Scale; HC—hormonal contraception; MDD—major depression disorder; QIDS—16-Item Quick Inventory of Depressive Symptomatology; OR—odds ratio; CI—confidence interval; PCOS—polycystic ovary syndrome; Ais—Aromatase Inhibitors; MD—major depression; PHQ-9—Patient Health Questionnaire-9; GnRHa—gonadotropin-releasing hormone agosnists; HAM-A/—Hamilton Scales for Anxiety/Depression; VAS-A/D—Visual Analogue Scale for Anxiety/Depression.

**Table 3 brainsci-10-00384-t003:** Associations between infertility and the predisposition for upcoming health issues.

Number of Patients	Main Observations	Reference
1000 patients (couples)	After the completion of the FPI-derived questionnaire prior to the beginning of the procedure, CFA has been partially validated. A post hoc EFA has explained two associated factors and the invariance between genders regarding stress-related states	[107]
1146 infertile patients	Using GAD-7 in parallel with a simple and multiple logistic method of classification has revealed that generalised anxiety is common among infertile women, which was positively correlated with four specific indices	[108]
1506 infertile patients	In accordance with PHQ-9 scores and through a simple and multiple logistic method of classification, it has been concluded that depression is predominant in women, a series of variables being positively correlated with its triggering	[109]
511 infertile women and 1017 controls	Based on a personalised version of SF-36, it has been concluded that the relatives have a significant impact upon the decision making, more precisely towards divorce, remarriage or adoption, independently of the social degree	[110]
1620 infertile women	While women recorded low scores in FertiQol in three specific subscales and high scores in SCREENIVF, this indicates that women present a high risk for developing emotional problems in contrast with their partners, both during and after the procedure	[111]
2180 patients from which 1049 are men and 1131 women	After the measurement of the severity of depression-like symptoms and infertility distress with MHI-5 and COMPI Fertility Problem Stress Scales, the predisposition for neuropsychiatric disorders have been almost three times higher, which was directly correlated with infertility-related distress in both groups	[112]
338 infertile patients and 1953 controls	Based on CIDI, BDI and GHQ-12, approximately 29% of the patients who had experienced infertility, especially women had increased risks for PDD and anxiety compared with controls. Those who have a child were more prone to panic disorder, while in men there was a reduction in the QOL	[113]
1468 infertile men and 942 controls	IIEF-5, PEDT and IELT, concomitant with SAS and SDS have revealed that compared with controls, the incidence of PE and ED has been significantly higher for the infertile patients, similar for anxiety and depression. Negative associations have been noted in IELT and IIEF-5 for anxiety and depression	[114]
2783 men	In total, 1750 men completed the Androgen Deficiency in the Aging Male (ADAM) and the Sexual Health Inventory for Men (SHIM) questionnaires. Through a multivariable logistic model, positive correlations between the prevalence for ED and the results obtained in ADAM have been noted. After the serum measurements of a series of biomarkers, no associations between T values with the symptoms of ED or TDS have been reported	[115]
5936 infertile women	1031 women who had never sought specialised advice for their infertility problem showed higher odds for depression, ovulation and metabolic disorders. Even though 728 of them pursued a treatment, 303 displayed increased chances to develop depression, tumours, menstrual disorders or infections at the level of urinary tract	[116]
6567 women with or without a history of IVF	With a median of seventeen years follow-up and by using a multivariate predictor, 411 women from the cohort have been admitted with mental diagnostics within the hospital, from which 93 had previously pursued IVF and 318 did not	[117]
9175 infertile women and 9175 controls	Women who had previously received treatment were less likely to be hospitalised for mental disorders or substance abuse/intoxication compared to controls. This risk was statistically significant, similar for hospitalisation during a decade post-treatment follow-up, but with exceptions in two indices. Furthermore, those who had given birth were less likely predisposed for anxiety, depression and substance abuse/intoxication in contrast with those who did not, the percentages regarding hospitalisations being identical between women who did not have a baby and controls	[118]
13,027 infertile men and 23,860 controls	It has been established that infertile men are predisposed to metabolic and cardiovascular disorders and substance abuse in contrast with those who underwent testing only or were vasectomised	[119]
98,320 women	When a pregnancy failed, women were at increased risk for psychiatric disorders and substance abuse compared with the women who gave birth after the infertility evaluation. No significant differences have been noted regarding the prevalence of anxiety, eating disorders or OCD	[120]
HUNT 2006–2008 *n* = 9200 womenHUNT 1 and HUNT 2 *n* = 5873 sub-fertile women and HUNT 2 *n* = 12,987 women	North-Trøndelag Health Study has been one of the biggest cross-sectional population-based studies designed to determine the predisposition of CNS disorders. Nevertheless, the results obtained are antithetical. No conclusive evidence has been found in two of them to link the incidence of infertility with the common mental health disorders, but the third confirms the causality	[121,122,123]

PFI—Fertility Problem Inventory; CFA—Confirmatory Factor Analysis; EFA—Exploratory Factor Analysis; GAD-7—Generalized Anxiety Disorder-7; PHQ-9—Patient Health Questionnaire-9; SF-36—The Short Form (36) Health Survey; FertiQol—The Fertility Quality of Life Questionnaire; MHI-5—Mental Health Inventory 5; CIDI—Composite International Diagnostic Interview; BDI—Beck Depression Inventory; GHQ-12—The 12-item General Health Questionnaire; PDD—Persistent Depressive Disorder; QOL—Quality Of Life; PE—Premature ejaculation; ED—Erectile dysfunction; PEDT—PE Diagnostic Tool; IELT—Intravaginal Ejaculatory Latency Time; IIEF-5—International Index of Erectile Function; SAS—Self-rating Anxiety Scale; SDS—Self-rating Depression Scale; IVF—In Vitro Fertilisation; OCD—Obsessive–Compulsive Disorder; CNS—Central Nervous System.

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
