# Peer review of "Mini-Review on the Possible Interconnections between the Gut-Brain Axis and the Infertility-Related Neuropsychiatric Comorbidities"

_brainsci, 2020, doi:10.3390/brainsci10060384_

Round 1
Reviewer 1 Report
Recension of manuscript No. brainsci-816990: „ Mini-review on the possible interconnections between the gut-brain axis and the infertility-related neuropsychiatric comorbidities, written by Gabriela Simionescu , Ovidiu-Dumitru Ilie , Alin Ciobica , Bogdan Doroftei , Radu Maftei , Delia Grab , Emil Anton”, which will be published in Brain Sciences.
The structure of manuscript is in keeping with the common required criteria. Both the gut-brain axis (GBA) and the hypothalamic–pituitary–adrenal (HPA) axis remain to date an intriguing and still obscure network considering its interdisciplinary nature and due to the fact that is in strong communication with other systems of organs.
Based on the aspects presented in this manuscript, it can be accepted that in fact, the gut microflora has a profound and complex influences on the psychological profile of each individual. Moreover, this integrative component not only constitute a bridge for upcoming CNS disorders, but also could provoke gastrointestinal deficiencies in an early stage. Regarding the infertility medication and the overall disease, this topic remains debatable, mainly because the number of studies is limited, but it is certain that it could disrupt the integrity of the GBA-HPA axes. However, a dysbacteriosis occured not only gradually alter the homeostasis, but amplify the chances to block entirely the effect of any infertility drug. On the other hand, the techniques dedicated to the restoration of the microbial communities have undergone a fulminant ascension lately, but like any therapy, some disorders have been omitted for unknown reasons, which is why additional studies are required.
Work is clearly legible, brings summarizes new knowledge. The citations are actual and their format respect usual standards. The conclusion reflects the author´s results and these can be accepted.
Author Response
Dear Reviewer #1,
Thank you very much for the positive feedback, interest, and time spent reviewing our manuscript. We appreciate the valuable comments which have helped us to create a much more appropriate mini-review and, therefore, to provide a clearer perspective. A part of our paper has been re-evaluated and redesigned by a team from the United Kingdom (Dr. Jack McKenna, Dr. Ioannis Mavroudis, and Dr. Nitasha Dhunna). Based on their field of expertise and previous experience, and even if the messages we wanted to convey to the reader have retained their meaning, word(s) or even entire sentences can no longer be found in the latest version. It is a measure we have resorted to because of the limited time.
We would like to thank you again very much.
Kind regards,
Bogdan Doroftei
Reviewer 2 Report
In the current review, Simionescu et al. highlight the recent updates on the relation between the Gut-brain Axis and The Infertility-Related Neuropsychiatric Comorbidities.
It summerizes the recent studies on the role of gut-brain in the modulation of infertility and related neuropsychiatric issues.
Authors have arranged a lot of recent work in tabular format, however, not clearly writing science they want to convey.
There are a number of errors in grammar which make it hard to understand.
For example, line 175. ''How can be modulated the HPA axis?''
line 218, ''Correlations between infertility-related HPA axis disturbance and the neuropsychiatric ones''
line 219-243, what exactly authors are expressing??? very hard to follow. There are number of similar errors throughout the paper.
Except tables, everything is very hard to follow. Please re-write in a simple language format.
To add some more examples, line 39, 53, 66..please write in a simple language.
What is line 71 explaining.??? completely confusing.
What are authors trying to tell about ''Integrative Human Microbiome Project (iHMP)'' in this context.
Author Response
Dear Reviewer #2,
Thank you very much for the positive feedback, interest, and time spent reviewing our manuscript. We appreciate the valuable comments which have helped us to create a much more appropriate mini-review and, therefore, to provide a clearer perspective. A part of our paper has been re-evaluated and redesigned by a team from the United Kingdom (Dr. Jack McKenna, Dr. Ioannis Mavroudis, and Dr. Nitasha Dhunna). Based on their field of expertise and previous experience, and even if the messages we wanted to convey to the reader have retained their meaning, word(s) or even entire sentences can no longer be found in the latest version. It is a measure we have resorted to because of the limited time.
Here is a list of all the changes we made and our responses:
line 175. ''How can be modulated the HPA axis?''
New subtitle: “How is the HPA axis influenced?”
line 218, ''Correlations between infertility-related HPA axis disturbance and the neuropsychiatric ones''
New subtitle: “How infertility and associated drugs disrupt the HPA axis”.
line 219-243, what exactly authors are expressing??? very hard to follow. There are number of similar errors throughout the paper.
We have tried to highlight the profound antagonistic effect of infertility on any individual’s psychological profile. It seems that the related medication for infertility indeed possess a side-effect panel, and contributes as well to the disturbance of the HPA axis. In tandem with the awareness of the current “disease”, both medication and the psychological profile amplifies exponentially the risk for various neurological disorders.
Except tables, everything is very hard to follow. Please re-write in a simple language format.
To add some more examples, line 39, 53, 66..please write in a simple language.
Dear Reviewer, a part of our manuscript has been re-evaluated and redesigned by a team from the United Kingdom.
What is line 71 explaining.??? completely confusing.
Line 71 refers to the multitude of factors that shape the human gastrointestinal microflora, and how in a collaborative manner provoke a dysbiosis along the digestive tract, from the stomach, until the colon and appendix.
What are authors trying to tell about ''Integrative Human Microbiome Project (iHMP)'' in this context.
When it was launched the so-called Human Microbiome Project, the main objectives were to develop libraries with reference sequences and technologies for exploring the variability of the human microbiome (perspective strictly scientific). However, the field being obscure, it was launched the second phase or Integrative Human Microbiome Project (it included both clinical and research perspective).
We would like to thank you again very much!
Kind regards,
Bogdan Doroftei
Reviewer 3 Report
Dear Authors,
Congratulations on an excellent manuscript, which I am delighted to recommend for publication in its present form.
With kind regards,
Reviewer

Author Response
Dear Reviewer #3,
Thank you very much for the positive feedback, interest, and time spent reviewing our manuscript. We appreciate the valuable comments which have helped us to create a much more appropriate mini-review and, therefore, to provide a clearer perspective. A part of our paper has been re-evaluated and redesigned by a team from the United Kingdom (Dr. Jack McKenna, Dr. Ioannis Mavroudis, and Dr. Nitasha Dhunna). Based on their field of expertise and previous experience, and even if the messages we wanted to convey to the reader have retained their meaning, word(s) or even entire sentences can no longer be found in the latest version. It is a measure we have resorted to because of the limited time.
We would like to thank you again very much.
Kind regards,
Bogdan Doroftei
Round 2
Reviewer 2 Report
No more comments.